# Simple Stepwise Approach to Differentiate Cyst-Like Soft-Tissue Masses by Using Time-Resolved Magnetic Resonance Angiography

**DOI:** 10.3390/diagnostics10121094

**Published:** 2020-12-15

**Authors:** Ying-Chieh Lai, Yu-Hsiang Juan, Shu-Hang Ng, Tzu-Chin Lo, Wen-Yu Chuang, Chun-Chieh Chen, Chi-Ting Liau, Gigin Lin, Yu-Jr Lin, Yu-Ching Lin

**Affiliations:** 1Department of Medical Imaging and Intervention, Chang Gung Memorial Hospital at Linkou, Institute for Radiological Research, Chang Gung University, Taoyuan 333, Taiwan; cappolya@gmail.com (Y.-C.L.); jonat126@yahoo.com.tw (Y.-H.J.); shuhangng@gmail.com (S.-H.N.); m1123@cgmh.org.tw (T.-C.L.); giginlin@cgmh.org.tw (G.L.); 2Department of Medical Imaging and Intervention, Chang Gung Memorial Hospital at Taoyuan, Institute for Radiological Research, Chang Gung University, Taoyuan 333, Taiwan; 3Department of Pathology, Chang Gung Memorial Hospital at Linkou, Chang Gung University, Taoyuan 333, Taiwan; chuang.taiwan@gmail.com; 4Department of Orthopaedic Surgery, Bone and Joint Research Center, Chang Gung Memorial Hospital at Linkou, Chang Gung University, Taoyuan 333, Taiwan; ms8614@gmail.com; 5Division of Hematology-Oncology, Department of Internal Medicine, Chang Gung Memorial Hospital at Linkou, Chang Gung University, Taoyuan 333, Taiwan; gerson@ms2.hinet.net; 6Research Services Center for Health Information, Chang Gung University, Taoyuan 333, Taiwan; doublelin15@gmail.com; 7Department of Medical Imaging and Intervention, Chang Gung Memorial Hospital at Keelung, Institute for Radiological Research, Chang Gung University, Keelung 204, Taiwan

**Keywords:** cyst, soft-tissue, tumor, magnetic resonance, angiography

## Abstract

This retrospective study aimed to differentiate cyst-like musculoskeletal soft-tissue masses by using time-resolved magnetic resonance angiography (MRA). During May 2015 to November 2019, patients with cyst-like soft-tissue masses examined through contrast-enhanced MRI followed by histologic diagnosis were included. The masses were classified into vascular lesions, solid lesions, and true cysts. Size, T1 hyperintensity, T2 composition, perilesional edema, time-resolved MRA, and static internal enhancement were assessed. The time-resolved MRA manifestations were classified into vascular pooling, solid stain, and occult lesion. Imaging predictors for each type of mass were identified through logistic regression and were used to develop a diagnostic flowchart. A total of 80 patients (47 men; median age, 42 years) were included, with 22 vascular lesions, 38 solid lesions, and 20 true cysts. The T2 composition, time-resolved MRA, and static internal enhancement were significantly different among the masses. Vascular pooling on time-resolved MRA was the sole predictor of vascular lesions (odds ratio = 722.0, *p* < 0.001). Solid stain on time-resolved MRA was the sole predictor of solid lesions (odds ratio = 73.6, *p* < 0.001). Occult lesion on time-resolved MRA (odds ratio = 7.4, *p* = 0.001) and absence of static internal enhancement (odds ratio = 80.0, *p* < 0.001) both predicted true cysts, while the latter was the sole predictor of true cysts after multivariate analysis. A diagnostic flowchart based on time-resolved MRA correctly classified 89% of the masses. In conclusion, time-resolved MRA accurately differentiates cyst-like soft-tissue masses and provides guidance for management.

## 1. Introduction

A cyst-like soft-tissue mass is defined as a lesion having bright signal intensity, equal to that of fluid, on T2-weighted or fluid-sensitive MRI sequences [1,2]. These lesions include true cysts and cyst-like masses (vascular lesions or solid lesions), and they could substantially overlap in images [1,3,4]. True cysts, such as ganglions or synovial cysts, may have a complex appearance and mimic solid lesions due to hemorrhage or infection [1]. Vascular lesions, such as hemangiomas or vascular malformations, may be heterogeneously bright on T2-weighted images because of mixed slow and rapid blood flow within lesions [5,6,7]. Solid lesions may appear cystic due to central necrosis, intratumoral edema, or fluid-containing stroma [1,8]. Thus, the definite diagnosis of a cyst-like soft-tissue mass based on imaging alone can be challenging, and a histopathology analysis is often warranted to confirm the diagnosis. However, a percutaneous biopsy of hemangioma or vascular malformation may lead to uncontrollable hemorrhage. Moreover, a biopsy of vascular lesions usually yields many blood products with insufficient tissue, making it difficult to obtain a histologic diagnosis [6]. Notably, the clinical management of cyst-like soft-tissue masses can be substantially different. Vascular lesions can be treated with microinvasive embolotherapy or sclerotherapy. True cysts can be conservatively treated, whereas solid lesions, especially malignancy, should be accurately recognized for early treatment. Thus, developing a practical imaging approach is crucial for clinical management.

Gray-scale ultrasonography is commonly used for initial assessment of musculoskeletal soft-tissue masses, and Doppler ultrasonography provides vascular information of the lesion [9]. However, ultrasonography is operator dependent and has limited field of view and imaging depth. Because vascular lesions can mimic other soft-tissue masses, catheter-directed digital subtracted angiography (DSA) may be required for a definite diagnosis [10,11,12]. DSA provides dynamic vascular imaging with a high temporal and spatial resolution; however, it is an invasive and relatively time-consuming procedure. In recent years, noninvasive time-resolved magnetic resonance angiography (MRA) is commonly used as an alternative imaging modality to DSA. Time-resolved MRA is a series of images obtained at multiple time points to display the dynamic flow of contrast medium through the blood vessels. It allows rapid acquisition of three-dimensional images which provide detailed vascular hemodynamic and anatomic information [13,14,15]. Investigators have demonstrated a high correlation between time-resolved MRA and DSA in assessing peripheral arterial occlusive disease [16]. Furthermore, time-resolved MRA has shown promising results in delineating the vascular architecture and flow pattern for musculoskeletal tumors and vascular malformations [17,18,19]. However, the ability of time-resolved MRA to differentiate cyst-like soft-tissue masses has not been fully explored in the literature.

The aim of this study is to explore the diagnostic potential of time-resolved MRA and to develop a practical approach to differentiate cyst-like soft-tissue masses.

## 2. Materials and Methods

### 2.1. Study Design

The institutional review board approved the protocol of this retrospective study (IRB number: 201901520B0; 16 October 2019), and a waiver of consent was obtained. On the basis of histologic diagnosis, cyst-like soft-tissue masses were classified into a vascular lesion, solid lesion (including benign and malignant solid lesion), and true cyst. Imaging features were analyzed to identify imaging predictors for differentiating these masses and to design a stepwise diagnostic flowchart for cyst-like soft-tissue masses. Interobserver agreements were assessed.

### 2.2. Patients

Between May 2015 and November 2019, a total of 450 consecutive patients underwent contrast-enhanced MRI for symptomatic musculoskeletal masses. In this study, we included the masses that (a) presented cystic appearance on T2-weighted images [2]; (b) had cystic component representing more than two-third of the entire lesion; and (c) located mainly in subcutaneous or muscle tissues. We excluded masses that were not histologically diagnosed. Of the scanned patients, 145 patients met the inclusion criteria of this study; however, 65 patients were excluded as they lacked a histologic diagnosis. Finally, 80 patients were included in this study.

### 2.3. MRI Protocol

All MRI examinations were performed on 3-Tesla MRI scanner (Magnetom Skyra; Siemens Healthcare Sector, Erlangen, Germany). The institutional MRI protocol for evaluating the musculoskeletal tumor consisted of T2-weighted fast spine echo without and with fat suppression (repetition time/echo time, 3220–7500/44–86 ms), precontrast T1-weighted fast spine echo (437–776/7.8–20 ms), time-resolved MRA with interleaved stochastic trajectories (TWIST), and static postcontrast T1-weighted fast spine echo (490–855/8.2–17 ms). A time-resolved MRA sequence was acquired using the following parameters: 2.4–3.6/1.0–1.42 ms; flip angle, 20°; rectangular field of view, 349 by 399 mm^2^; matrix, 384 by 302; and acceleration factor, 3. In total, 0.2 mmol/kg bolus of contrast medium (Magnevist [gadopentetate dimeglumine], Bayer Healthcare, Wayne, NJ, USA) was injected at a rate of 2 mL/s, followed by a 15 mL saline flush by using an automated power injector. The scanning was initiated manually 10 s after contrast medium injection. For each time-resolved MRA sequence, 33 dynamic-phase scanning images with a temporal resolution of 2.93 s were obtained for covering arterial and venous phases. The scanning time was 12.4 s for the reference image and the last phase and 2.93 s for each dynamic phase, resulting in a total scanning time of 118 s. The readers routinely evaluated time-resolved MRA based on the maximum intensity projections images in three orthogonal planes; the three-dimensional source images at each time point were only used for trouble shooting.

### 2.4. Imaging Features

The following imaging features were analyzed: (a) size, average of the diameter in three planes; (b) T1 hyperintensity, defined as signal intensity higher than that of the muscle on T1-weighted images, which may correspond to fat, methemoglobin, proteinaceous material, or melanin [2]; (c) T2 composition, graded as homogeneous or heterogeneous on T2-weighted images; (d) perilesional edema, defined as a region of high signal intensity surrounding the lesion on fat-suppressed T2-weighted images; and (e) time-resolved MRA manifestations, subjectively classified into three different patterns—vascular pooling, solid stain, and occult lesion. Vascular pooling was defined as dilated vascular channels or cavities that have persistent contrast medium puddling even in the late dynamic phase (Figure 1), solid stain was defined as tumor-like contrast medium blush (Figure 2), and occult lesion was assigned if the lesion was not discernible on time-resolved MRA (Figure 3); (f) internal enhancement on static-enhanced MRI, subjectively stratified into three degrees—marked, mild–moderate, and absent. The enhancement equal to that of the slow-flow blood vessels was defined as marked enhancement, and the enhancement between marked and absent enhancement was considered as mild–moderate enhancement. Two radiologists (Y.-C. Lai and Y.-C. Lin, with 5 and 15 years of musculoskeletal imaging experience, respectively) independently reviewed the images, and the final results were based on a consensus in case of discrepancy. The imaging readers were blinded to histologic diagnoses during image interpretations.

### 2.5. Histopathology Analysis

A pathologist (W.-Y.C., with 20 years of musculoskeletal pathology experience) examined specimens and assigned the histologic diagnosis according to the World Health Organization classification of tumors of the soft-tissue and bone [3].

### 2.6. Statistical Analysis

All data were analyzed using SPSS version 25 (Armonk, NY, USA). The Wilcoxon rank sum test was used to compare the age between male and female patients. The Kruskal–Wallis test was used to compare the size. The χ^2^ or Fisher’s exact test was used to examine any significant difference in the frequency distribution of categorical imaging features. Imaging features were entered into univariate and multivariate logistic regression with stepwise procedure. The interobserver agreement for the imaging feature was calculated using the *κ* statistic. A *κ* of 0.81–1.0 indicated excellent agreement, 0.61–0.80 indicated good agreement, 0.41–0.60 indicated moderate agreement, 0.21–0.40 indicated fair agreement, and 0–0.20 indicated only slight agreement. Two-tailed *p* < 0.05 was considered statistically significant.

## 3. Results

The study patients consisted of 47 men (median age, 43 years; range, 2–77 years) and 33 women (median age, 41 years; range, 11–78 years); there was no significant difference in the ages of male and female patients (*p* = 0.86). Cyst-like soft-tissue masses were located in the lower limb (30 of 80, 38%), upper limb (41 of 80, 51%), and body (9 of 80, 11%). A total of 22 vascular lesions, 38 solid lesions, and 20 true cysts were analyzed (Table 1). The vascular lesions consisted of 20 hemangiomas and 2 lymphatic malformations; the solid lesions consisted of 22 benign solid lesions and 16 malignant solid lesions.

The frequency distribution of imaging features among the cyst-like soft-tissue masses is displayed in Table 2. There was no significant difference in the size among the masses (*p* = 0.32). The T2 composition, time-resolved MRA patterns, and static internal enhancement were significantly different among the masses (*p* = 0.01, < 0.001, and <0.001, respectively). On time-resolved MRA, 19 of the 20 hemangiomas presented as vascular pooling, and both lymphatic malformations were occult. Most of the solid lesions showed solid stain (31 of 38), whereas six solid lesions were occult. True cysts were most likely to be occult (13 of 20), whereas seven true cysts showed solid stain. The imaging predictors of each type of mass are detailed in Table 3. Vascular pooling on time-resolved MRA was the sole predictor of vascular lesions (odds ratio = 722.0, *p* < 0.001). Solid stain on time-resolved MRA was the sole predictor of solid lesions (odds ratio = 73.6, *p* < 0.001). Occult lesion on time-resolved MRA (odds ratio = 7.4, *p* = 0.001) and absence of static internal enhancement (odds ratio = 80.0, *p* < 0.001) both predicted true cysts, while the latter was the sole predictor of true cysts after multivariate analysis. The accuracy of time-resolved MRA and static internal enhancement in evaluating cyst-like soft-tissue masses are compared in Table 4. Time-resolved MRA was more accurate than static enhancement in identifying vascular lesions and solid lesions (accuracy = 95% [76/80] and 81% [65/80], respectively), while static enhancement was more accurate than time-resolved MRA in identifying true cysts (accuracy = 93% [74/80]). Of the 21 occult lesions on time-resolved MRA, 13 occult lesions were confirmed to have no internal enhancement on static-enhanced MRI, and all of them were true cysts (Figure 3). However, 8 occult lesions showed internal enhancement on static-enhanced MRI—6 lesions having central enhancement were solid lesions and 2 lesions demonstrating septal enhancement were lymphatic malformations.

A diagnostic flowchart was developed for cyst-like soft-tissue masses on the basis of time-resolved MRA patterns, as demonstrated in Figure 4. In the flowchart, static enhancement was simplified, graded as presence or absence of static internal enhancement. According to this flowchart, 71 of the 80 included masses (89%) were correctly classified. Nearly all vascular lesions (21 of 22, 95%) and solid lesions (37 of 38, 97%) could be identified through the flowchart. Seven true cysts (7 of 20, 35%) presented as solid stain on time-resolved MRA and were miscategorized as solid lesions. Interobserver agreements were good, with *κ* of 0.745 for time-resolved MRA patterns and 0.691 for static internal enhancement.

## 4. Discussion

The integration of noninvasive time-resolved magnetic resonance angiography before static contrast-enhanced images provides new insights into lesion characterization for cyst-like soft-tissue masses. A study by Wu et al. emphasized static internal enhancement for differentiating between true cysts and cyst-like tumors [2]. Another study by Bermejo et al. used heterogeneity, wall thickness, location, and static internal enhancement for evaluating simple and complicated true cysts as well as solid lesions [1]. Both studies focused on the imaging features on conventional MRI. Our study provided an additional method for differentiating cyst-like masses into vascular lesions, solid lesions, and true cysts by using time-resolved MRA. The use of proposed flowchart resulted in accurate diagnosis of the vascular lesions and the solid lesions, which provided valuable clinical information to determine whether percutaneous or surgical intervention should be performed. Although the true cysts mimicked solid lesions on time-resolved MRA, these lesions can undergo biopsy or surgical resection with a relatively low risk of procedure-related hemorrhage.

Vascular soft-tissue lesions with a cystic appearance were predominantly hemangiomas. The literature has described imaging features of soft-tissue hemangiomas, which include bright signal intensity on T2-weighted images, lobulation and septation, and central low-signal-intensity dots [7,20,21,22]. However, hemangiomas may not present all these features on conventional MRI. Hemangiomas generally had greater static internal enhancement compared with malignant soft-tissue masses, but the difference was not statistically significant when objectively measured [20]. Thus, biopsy is sometimes necessary to achieve a definite diagnosis. Our study included hemangiomas with atypical features on conventional MRI that were diagnosed by histopathology analysis. For these lesions, time-resolved MRA provided accurate differentiation of hemangiomas from other cyst-like soft-tissue masses. Histologically, hemangiomas consist of discrete vascular channels and blood-filled spaces with interspersed nonvascular components such as fat, muscle, myxoid, or fibrous tissue [23]. This explains the unique appearance of hemangiomas on either DSA or time-resolved MRA, that is, gradual filling of small vessels that run into wide vascular cavities and remain there for several seconds [11,22]. By contrast, solid stain is more homogeneous and tumor-like on time-resolved MRA presumably due to a vascular-rich stroma of solid lesions. The observers in our study easily differentiated between these two different patterns, regardless of their experience in MRI interpretation. Moreover, interobserver agreement was good, suggesting that this is a simple and practical approach. Notably, true cysts may appear as solid stain on time-resolved MRA if they are complicated with infection or rupture, because the outer rim enhancement is superimposed with the non-enhancing inner potion on maximum intensity projections images.

We developed a diagnostic flowchart based on time-resolved MRA patterns. The accurate differentiation between hemangiomas and solid lesions by using this approach is valuable in clinical practice. Because differentiating cyst-like soft-tissue masses through imaging may be challenging, a histopathology analysis through biopsy is often required for a definite diagnosis. However, biopsy of a hemangioma is associated with the risk of hemorrhage and low diagnostic yield [6]. Using this flowchart, hemangiomas can be confidently differentiated from solid lesions to avoid unnecessary and potentially risky biopsy. A systemic review by Kim et al. also highlighted the use of time-resolved MRA to facilitate diagnosis of vascular lesions in pediatric patients [24].

If the mass is occult on time-resolved MRA, static-enhanced MRI should be evaluated before establishing a diagnosis of a true cyst. Although true cysts could be predicted through occult lesions on time-resolved MRA, occult lesions were found to be vascular lesions or solid lesions in our study. This can be attributed to the following reasons. First, a subtle enhancement of the mass may not be clearly depicted owing to the reduced spatial resolution of time-resolved MRA due to the dynamic nature of the sequence [15]. Second, a small enhancement of mass may be masked by overlapping vessels. Lastly, in patients with a compromised hemodynamic status or severe vascular occlusive diseases, time-resolved MRA may not provide adequate temporal coverage for venous phase, and a mass with delayed enhancement may not be captured. Thus, further evaluation of occult lesions on static-enhanced MRI is warranted. True cysts show completely absent or only a thin rim enhancement, whereas vascular lesions or solid lesions demonstrate internal enhancement [2]. Of note, lymphatic malformation, a vascular lesion defined by the World Health Organization classification, is angiographically occult because it comprises only dilated channels or sinuses connected to the lymphatic system. Lymphatic malformation relies on static-enhanced MRI to characterize the vascularized and thus enhancing septa [25].

Our study has limitations. First, uncommon cystic soft-tissue lesions, such as hydatid cysts, were not included, and this could not be avoided because this study was performed at a single center with a limited sample size. Second, our study did not include any malignant vascular lesion. However, angiosarcoma is a rare entity, accounting for less than 1% of all sarcomas, and they usually present as a solid mass [26]. Third, this study is retrospective, and only pretreatment lesions were included in this study. It is unclear whether the results of this study can be used to evaluate recurrent lesions with a cystic appearance. Lastly, this study did not differentiate between benign and malignant solid lesions because only a small number of sarcomas were analyzed. Benign soft-tissue lesions are much more frequently encountered than sarcomas in practice. Thus, a multicenter study with a large cohort is warranted to provide a more comprehensive evaluation of the cyst-like soft-tissue masses.

In conclusion, time-resolved magnetic resonance angiography before static-enhanced MRI provides differentiation of cyst-like soft-tissue masses into vascular lesions, solid lesions, and true cysts. We recommend incorporating time-resolved MRA into the MRI evaluation of cyst-like soft-tissue masses to avoid unnecessary and potentially risky biopsy.

## Figures and Tables

**Figure 1 diagnostics-10-01094-f001:**
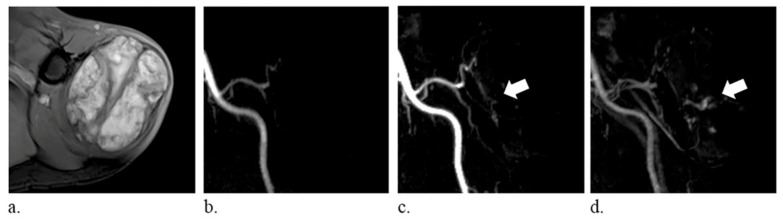
A 74-year-old man with hemangioma of the left shoulder. (**a**) fat-suppressed T2-weighted axial images (repetition time/echo time, 6060/62 ms) show a heterogeneously hyperintense mass. (**b**–**d**) time-resolved magnetic resonance angiography in the coronal plane demonstrates persistent vascular pooling within the lesion (arrows).

**Figure 2 diagnostics-10-01094-f002:**
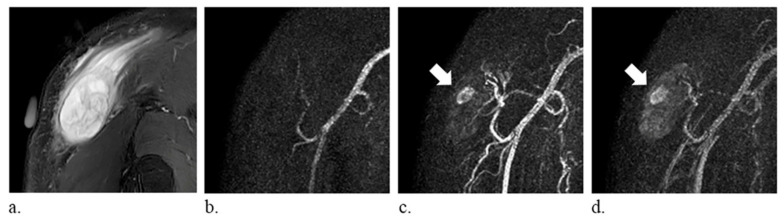
A 59-year-old man with myxofibrosarcoma of the right shoulder. (**a**) fat-suppressed T2-weighted coronal images (repetition time/echo time, 4000/75 ms) show a heterogeneously hyperintense mass (arrow). (**b**–**d**) time-resolved magnetic resonance angiography in the coronal plane demonstrates tumor-like solid stain (arrow).

**Figure 3 diagnostics-10-01094-f003:**
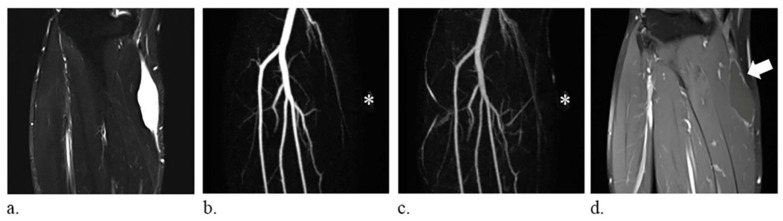
A 59-year-old man with a synovial cyst of the right leg. (**a**) fat-suppressed T2-weighted coronal images (repetition time/echo time, 6780/74 ms) show a homogeneously hyperintense mass. (**b**–**c**) the lesion, indicated with an *, is not discernible on the coronal images of time-resolved magnetic resonance angiography and therefore defined as occult lesion. (**d**) static-enhanced T1-weighted coronal images (500/9.1 ms) with fat suppression reveal absence of internal enhancement of the lesion (arrow).

**Figure 4 diagnostics-10-01094-f004:**
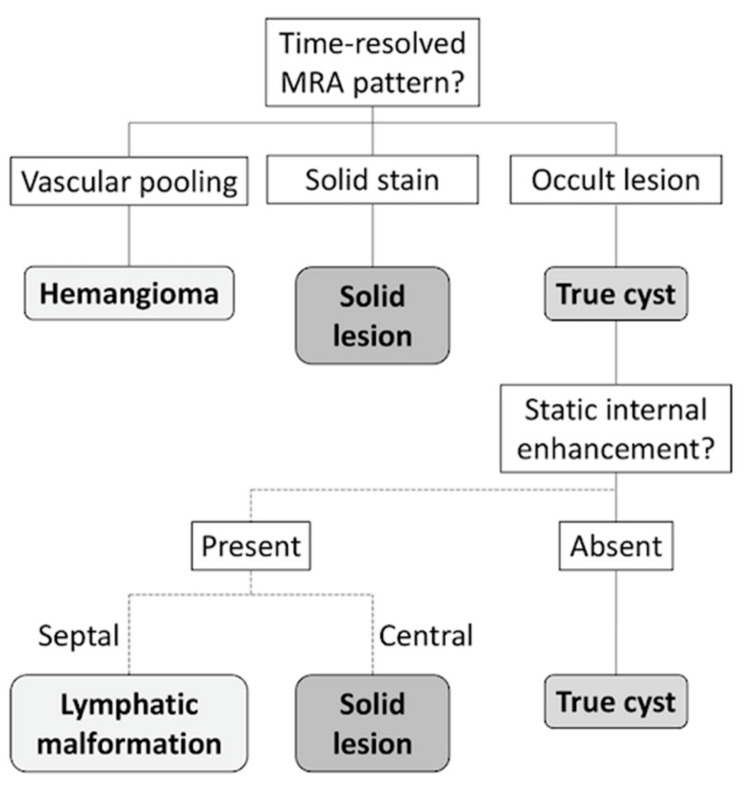
Flowchart for evaluation of cyst-like soft-tissue masses. Time-resolved MRA pattern of the mass was evaluated, and vascular pooling and solid stain on time-resolved MRA indicated hemangioma and solid lesion, respectively. If the mass is occult on time-resolved MRA, static-enhanced MRI should be evaluated before establishing the diagnosis of a true cyst.

**Table 1 diagnostics-10-01094-t001:** Histologic diagnosis of cystic-like soft-tissue masses.

Histologic Diagnosis	Number of Cases
Vascular lesion	-
Hemangioma	20
Lymphatic malformation	2
Benign solid lesion	-
Benign peripheral nerve sheath tumor	12
Glomus tumor/myopericytoma	8
Tenosynovial giant cell tumor	1
Fibromatosis	1
Malignant solid lesion	-
Infantile fibrosarcoma	1
Myxofibrosarcoma	6
Malignant giant cell tumor	1
Malignant peripheral nerve sheath tumor	2
Undifferentiated pleomorphic sarcoma	1
Squamous cell carcinoma	1
Sarcoma of uncertain differentiation	2
Myxoid liposarcoma	2
True cyst	-
Ganglion/synovial cyst	10
Abscess	6
Epidermoid cyst	2
Hematoma	2

**Table 2 diagnostics-10-01094-t002:** Distribution and comparison of imaging features in cystic-like soft-tissue masses.

Imaging Feature	Vascular Lesion (*n* = 22)	Solid Lesion (*n* = 38)	True Cyst (*n* = 20)	*p* Value
Size (cm)				0.32
Median	2.9	3.0	3.9	
Range	0.7–10.8	0.3–17.2	0.7–10.3	
T1 hyperintensity				0.77
Absence	9	19	10	
Presence	13	19	10	
T2 composition				0.01 *
Homogeneous	1	11	9	
Heterogeneous	21	27	11	
Perilesional edema				0.10
Absence	16	20	8	
Presence	6	18	12	
Time-resolved MRA				<0.001 *
Vascular pooling	19 †	1	0	
Solid stain	1	31	7	
Occult lesion	2	6	13	
Static enhancement				<0.001 *
Marked	15	28	2	
Mild–moderate	6 ‡	10	3	
Absent	1	0	15	

MRA = magnetic resonance angiography; * significant *p* value; † all vascular lesions showing vascular pooling were hemangiomas in this study; and ‡ two lymphatic malformations demonstrated mildly enhanced internal septa on static-enhanced MRI.

**Table 3 diagnostics-10-01094-t003:** Imaging predictors of cyst-like soft-tissue masses after univariate logistic regression.

		Vascular Lesion			Solid Lesion			True Cyst	
Variables	OR	95% CI	*p* Value	OR	95% CI	*p* Value	OR	95% CI	*p* Value
Size (cm)	1.0	0.8–1.2	0.984	0.9	0.8–1.1	0.488	1.1	0.9–1.3	0.437
T1 hyperintensity	1.4	0.5–3.9	0.468	0.8	0.3–2.0	0.670	0.9	0.3–2.4	0.796
T2 heterogeneous	1.7	0.9–3.3	0.100	0.8	0.5–1.5	0.550	0.7	0.4–1.3	0.305
Perilesional edema	0.4	0.1–1.0	0.055	1.2	0.5–2.9	0.686	2.2	2.2–6.3	0.124
Time-resolved MRA									
Vascular pooling	722.0	42.8 to >999.9	<0.001 *		Ref		<0.1	<0.1–0.0	0.998
Solid stain		Ref		73.6	8.5–635.8	<0.001 *		Ref	
Occult lesion	4.0	0.3–47.0	0.270	7.6	0.8–70.2	0.074	7.4	2.2–24.7	0.001 *
Static enhancement									
Marked	7.5	0.9–62.3	0.062		Ref		0.2	<0.1–1.6	0.146
Mild–moderate	6.9	0.7–65.3	0.091	0.7	0.2–2.0	0.476		Ref	
Absent		Ref		<0.1	<0.1–0.0	0.998	80.0	7.5–856.0	<0.001 **

OR = odds ratio and CI = confidence interval; * significant *p* value; and ** sole independent predictor after multivariate logistic regression.

**Table 4 diagnostics-10-01094-t004:** Diagnostic accuracy of time-resolved MRA and static internal enhancement in the evaluation of cyst-like soft-tissue masses.

Type of Mass/Imaging Feature	Sensitivity (%)	Specificity (%)	PPV (%)	NPV (%)	Accuracy (%)
Vascular lesion					
Vascular pooling on MRA	86 (65–97; 19/22)	98 (91–100; 57/58)	95 (75–100; 57/60)	95 (86–99; 19/20)	95 (88–99; 76/80)
Marked enhancement	68 (45–86; 15/22)	48 (35–62; 28/58)	33 (20–49; 15/45)	80 (63–92; 28/35)	54 (42–65; 43/80)
Solid lesion					
Solid stain on MRA	82 (66–92; 31/38)	81 (66–91; 34/42)	79 (64–91; 31/39)	83 (68–93; 34/41)	81 (71–89; 65/80)
Mild–moderate enhancement	26 (13–43; 10/38)	79 (63–90; 33/42)	53 (29–76; 10/19)	54 (41–67; 33/61)	54 (42–65; 43/80)
True cyst					
Occult lesion on MRA	65 (41–85; 13/20)	87 (75–94; 52/60)	62 (38–82; 13/21)	88 (77–95; 52/59)	81 (71–89; 65/80)
Absent enhancement	75 (51–91; 15/20)	98 (91–100; 59/60)	94 (70–100; 15/16)	92 (83–97; 59/64)	93 (84–97; 74/80)

Data in parentheses are 95% confidence intervals and raw data; PPV = positive predictive value and NPV = negative predictive value.

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
