# Peer review of "Simple Stepwise Approach to Differentiate Cyst-Like Soft-Tissue Masses by Using Time-Resolved Magnetic Resonance Angiography"

_diagnostics, 2020, doi:10.3390/diagnostics10121094_

Round 1

Reviewer 1 Report

This is a retrospective analysis of the use of MRA in differentiating cyst-like soft tissue masses in musculoskeletal pathology. 

The field of musculoskeletal pathology is indeed a challenging one, but diagnostic tools are not limited. A vascular lesion is a diagnosis of exclusion. In my opinion, an MRA should be performed to differentiate malignant tumors from vascular lesions and this approach was discussed in this paper.   Being a topic of great interest, the approach of this paper is of great use.  Diagnosing true cystic lesions should not be a challenge in daily practice. But, comparing malign lesions vs vascular lesions, the results presented by this paper are promising.   Some specific observations:    Title: ....by using Time-resolved Magnetic Eesonance Angiography  maybe the authors meant: Resonance Angiography   Abstract: Should include the fact that the lesions were MSK soft tissue masses. The term neoplasm should be better defined. The reader needs to understand that both malignant and benign solid lesions were included. As a suggestion, the term neoplasm could be changed with solid benign lesions and solid malignant lesions.    Introduction appropriate   Methods  appropriate   Data analysis:  I would consider analyzing all parameters together and test their diagnosis value.    Discussion appropriate   Conclusion: appropriate   References: appropriate   Tables and Figures: appropriate    

Reviewer 2 Report

Abstract: The paragraph “In the multivariate analysis, time-resolved MRA was the sole predictor—vascular pooling, solid 35 stain, and occult lesion indicated vascular lesions, neoplasms, and true cysts, respectively (odds 36 ratio = 688.1, 9.5, and 7.0, respectively; accuracy = 95%, 81%, and 81%, respectively)” is difficult to understand.

Introduction: The authors should provide some information regarding time-resolved MRA for the unfamiliar readers. In my opinion, even this article is about MRI, some data about the diagnostic value of ultrasound should be added.

Materials and methods: Grammatical errors. For example: “the patients were screened”; the word “screened” should be replaced with “scanned”. Why T1 hyperintensity was analyzed?. The definition of the vascular pooling should be written in the text; also solid stain definition, in my opinion should be presented in the text, at the first mentioning of the term.

Results: In my opinion the accuracy of time-resolved MRA and static-enhanced MRI should be compared; adding a supplementary sequences will increase the time of examination, which may lead to the movement of the patient and compromising the quality of the examination. Also, in my opinion the results section should be simplified. For example, the data presented in the table 4 are also described in the results, the data being similar; I think that the authors should present these results in a single way (either table, either written in the text), otherwise the information is duplicated.

Discussion: The data “Our study provided an additional method for differentiating cyst-like masses into vascular lesions, neoplasms, and true cysts (accuracy = 96% (78 of 80), 88% (70 of 80), and 88% (70 of 80), respectively). The use of proposed flowchart resulted in the correct diagnosis of 96% (21 of 22) of the vascular lesions and 97% (37 of 38) of the neoplasms” are already presented in the result section. Grammatical errors. For example: “Our study has imitations”. Are there any other similar studies? If yes, the authors should compare the results with the data from the literature. Also, would worth it to have a multicentric study where bias is reduced and more important data may be added.

Figures: Figure 2: The paragraph “The solid stain is decribed as the lesion predominantly presenting as tumoral stain with progressive enhancement” is difficult to understand.

Round 2

Reviewer 2 Report

Dear authors, 

the manuscript is significantly improved and, in my opinion, can be published.